# Characterization of Postprandial Bile Acid Profiles and Glucose Metabolism in Cerebrotendinous Xanthomatosis

**DOI:** 10.3390/nu15214625

**Published:** 2023-10-31

**Authors:** Soumia Majait, Emma C. E. Meessen, Frederic Maxime Vaz, E. Marleen Kemper, Samuel van Nierop, Steven W. Olde Damink, Frank G. Schaap, Johannes A. Romijn, Max Nieuwdorp, Aad Verrips, Filip Krag Knop, Maarten R. Soeters

**Affiliations:** 1Department of Pharmacy and Clinical Pharmacology, Amsterdam UMC Location University of Amsterdam, 1105 AZ Amsterdam, The Netherlands; s.majait@amsterdamumc.nl; 2Department of Endocrinology and Metabolism, Amsterdam UMC Location University of Amsterdam, 1105 AZ Amsterdam, The Netherlands; e.c.meessen@amsterdamumc.nl (E.C.E.M.); samvannierop@gmail.com (S.v.N.); 3Department of Clinical Chemistry and Pediatrics, Amsterdam UMC Location University of Amsterdam, Laboratory Genetic Metabolic Diseases, Emma Children’s Hospital, 1105 AZ Amsterdam, The Netherlands; f.m.vaz@amsterdamumc.nl; 4Inborn Errors of Metabolism, Amsterdam Gastroenterology Endocrinology Metabolism, 1105 AZ Amsterdam, The Netherlands; 5Core Facility Metabolomics, Amsterdam UMC location University of Amsterdam, 1105 AZ Amsterdam, The Netherlands; 6Department of Experimental Vascular Medicine, Amsterdam University Medical Center, 1105 AZ Amsterdam, The Netherlands; e.m.kemper@amsterdamumc.nl; 7Department of Surgery, NUTRIM School of Nutrition and Translational Research in Metabolism, Maastricht University, 6229 ER Maastricht, The Netherlands; steven.oldedamink@maastrichtuniversity.nl (S.W.O.D.); frank.schaap@maastrichtuniversity.nl (F.G.S.); 8Department of General, Visceral and Transplantation Surgery, RWTH University Hospital Aachen, 52074 Aachen, Germany; 9Department of Internal Medicine, Amsterdam UMC Location University of Amsterdam, 1012 WX Amsterdam, The Netherlands; j.a.romijn@amsterdamumc.nl; 10Department of Vascular Medicine, Amsterdam UMC Location University of Amsterdam, 1105 AZ Amsterdam, The Netherlands; m.nieuwdorp@amsterdamumc.nl; 11Department of Neurology, Canisius Wilhelmina Hospital, 6532 SZ Nijmegen, The Netherlands; a.verrips@cwz.nl; 12Center for Clinical Metabolic Research, Gentofte Hospital, University of Copenhagen, 2900 Hellerup, Denmark; filip.krag.knop.01@regionh.dk; 13Steno Diabetes Center Copenhagen, 2730 Herlev, Denmark; 14Department of Clinical Medicine, Faculty of Health and Medical Sciences, University of Copenhagen, 1353 Copenhagen, Denmark

**Keywords:** cerebrotendinous xanthomatosis, mixed meal test, glucagon-like peptide 1, fibroblast growth factor 19

## Abstract

Cerebrotendinous xanthomatosis (CTX) is a rare inherited disease characterized by sterol 27-hydroxylase (CYP27A1) deficiency and, thus, a lack of bile acid synthesis with a marked accumulation of 7α-hydroxylated bile acid precursors. In addition to their renowned lipid-emulgating role, bile acids have been shown to stimulate secretion of the glucose-lowering and satiety-promoting gut hormone glucagon-like peptide 1 (GLP-1). In this paper, we examined postprandial bile acid, glucose, insulin, GLP-1 and fibroblast growth factor 19 (FGF19) plasma profiles in patients with CTX and matched healthy controls. Seven patients and seven age, gender and body mass index matched controls were included and subjected to a 4 h mixed meal test with regular blood sampling. CTX patients withdrew from chenodeoxycholic acid (CDCA) and statin therapy three weeks prior to the test. Postprandial levels of total bile acids were significantly lower in CTX patients and consisted of residual CDCA with low amounts of ursodeoxycholic acid (UDCA). The postprandial plasma glucose peak concentration occurred later in CTX patients compared to controls, and patients’ insulin levels remained elevated for a longer time. Postprandial GLP-1 levels were slightly higher in CTX subjects whereas postprandial FGF19 levels were lower in CTX subjects. This novel characterization of CTX patients reveals very low circulating bile acid levels and FGF19 levels, aberrant postprandial glucose and insulin profiles, and elevated postprandial GLP-1 responses.

## 1. Introduction

Cerebrotendinous xanthomatosis (CTX) is a genetic lipid storage disease caused by a mutation in the sterol 27-hydroxylase (CYP27A1) gene [1,2]. As CYP27A1 constitutes the rate-limiting enzyme of bile acid synthesis in the liver, this defect leads to reduced bile acid synthesis and accumulation of bile acid synthesis intermediates and their downstream metabolites including cholestanol. Under normal circumstances, the primary bile acids chenodeoxycholic acid (CDCA) and cholic acid (CA) are synthesized in the liver and are subsequently conjugated to taurine or glycine and stored in the gallbladder [3,4]. After excretion into the duodenum, bile acids are largely deconjugated and can be modified by intestinal bacteria to form secondary bile acids, such as the 7β-epimer of CDCA, ursodeoxycholic acid (UDCA). Both conjugated and unconjugated bile acids are reabsorbed in the small intestine and only a small portion is lost in the faeces. After small intestinal reuptake, bile acids are taken up by the liver and here, they are again conjugated to taurine or glycine, becoming ready for a new round of enterohepatic circulation [5]. In the liver, bile acids inhibit CYP7A1 constituting a negative feedback signal on bile acid synthesis [6].

The interrupted bile acid synthesis in CTX leads to a compensatory increase in the activity of cholesterol 7 alpha-hydroxylase (CYP7A1) [1] leading to a marked accumulation of 7α-hydroxylated bile acid precursors which are converted to cholestanol and characteristic bile alcohol glucuronides accumulating in extrahepatic tissues. As a consequence, CTX patients suffer from atherosclerosis, tendon xanthomas and neurological disease. In early childhood, patients typically show chronic diarrhoea, cataracts and failure to thrive [7,8]. Later in childhood, patients may show tendon xanthomas, low IQ or psychiatric illness. During adulthood, CTX may be present with spastic paresis, a drop in IQ, dementia, ataxia, dysarthria, seizures and/or peripheral neuropathy [9,10]. When the gene defect is known within a family, it enables earlier diagnosing and, thus, more effective treatment. CDCA has been used in the treatment of CTX due to its potent Farnesoid × receptor (FXR) agonistic effect [11]. FXR is an intranuclear bile acid receptor that suppresses CYP7A1 and, thus, plays a central role in the feedback control of bile acid synthesis and secretion [3]. After intestinal re-uptake, bile acids exert negative feedback on their own biosynthesis via FXR directly and via gut-derived FGF19 [3]. Additionally, patients are treated with statins to lower cholesterol levels and, thus, bile acid production [12]. Patients may suffer from fat-soluble vitamin deficiency and need supplementation therapy [13,14].

In addition to FXR, bile acids also work via the Takeda G protein-coupled receptor 5 (TGR5) [15,16] in mediating several metabolic effects of bile acids [17,18]. For instance, bile acid activation of TGR5 on entero-endocrine L cells has been demonstrated to control glucose metabolism via the release of glucagon-like peptide 1 (GLP-1) in these cells [19,20]. Whether disturbances in bile acid synthesis and the enterohepatic circulation of bile acids as seen in CTX affect GLP-1 release and glucose metabolism remains unknown.

In this study we performed mixed meal tests in CTX patients that were shortly withdrawn from CDCA and statin treatment and controls that were matched for age, sex and body mass index (BMI). We hypothesized that the aberrant CTX bile acid profile would have a different bile acid receptor activating signature than the bile acid profile of the matched controls with possible differential postprandial effects on GLP-1, insulin and glucose as well as FGF19.

## 2. Materials and Methods

### 2.1. Participants

Clinical care for CTX patients in the Netherlands is largely centered in the Department of Neurology, Canisius Wilhelmina Hospital, Nijmegen. At this clinic, we were able to recruit seven CTX patients who were able to provide oral and written informed consent (Table 1). Prior to study days (see below), CTX patients were asked to withhold their CDCA and statin therapy for three weeks. We included 7 healthy participants matched for age, sex and BMI. Exclusion criteria for both CTX and matched controls were any previous surgery or current diseases of the liver, biliary or gastrointestinal tract; ethanol abuse; weight loss or weight gain in excess of 10% of body weight in the 6 months prior to inclusion; use of any medication other than statins and CDCA or the use of a herbal supplement; fasting plasma glucose > 7.0 mmol/L; or average blood glucose levels/haemoglobin A1c (HbA1c) > 53 mmol/mol, creatinine > 120 μmol/L, or abnormal renal liver or thyroid function defined as >2 times the upper limit of the reference interval. This study was approved by the AMC Medical Ethics Committee and was conducted in accordance with the principles of the Declaration of Helsinki. This study was registered at the Dutch Trial Register (NL2595).

### 2.2. Study Design

On study days, participants were admitted at 07:30 a.m. to the Experimental and Clinical Research Unit in the Amsterdam University Medical Centers (location: Academic Medical Center) or the Department of Neurology in the Canisius Wilhelmina Hospital after an overnight fast. A cannula was inserted into an antecubital vein for blood sampling. This hand was kept in a heated hand box throughout the test to arterialize venous blood. At 09:30 a.m., baseline samples were taken for the determination of basal plasma bile acid, GLP-1, FGF19, glucose and insulin concentrations. At 10:00 a.m., participants consumed a standard liquid meal (Nutridrink, Nutricia, Zoetermeer, The Netherlands) equivalent to 25% of daily energy expenditure based on the Harris–Benedict equation. After ingestion of the liquid test meal, blood samples were obtained at 0, 15, 30, 45, 60, 75, 90, 120, 150, 180 and 240 min after the meal. Blood was collected into chilled tubes containing EDTA or heparin as anticoagulant on ice and immediately centrifuged, and plasma was subsequently stored at −20 °C until later batch analysis. For GLP-1 analysis, a dipeptidyl peptidase inhibitor (Ile-Pro-Ile, Sigma-Aldrich, St. Louis, MO, USA) was added to the tubes at 0.01 mg/mL, and plasma was stored at −80 °C for later batch analysis.

### 2.3. Laboratory Analyses

An ultra-high performance liquid chromatography (UPLC) tandem mass spectrometry (MS) method was used to detect plasma concentrations of CA, CDCA, deoxycholic acid (DCA) and UDCA in their conjugated and unconjugated forms in plasma samples [21]. Plasma glucose concentrations were analysed bedside using the glucose oxidation method (EKF Diagnostics, Barleben/Magdeburg, Germany). Insulin was determined in plasma samples on an IMMULITE 2000 system (Siemens Healthcare Diagnostics, Breda, The Netherlands). Plasma FGF19 was measured using an in-house-developed ELISA as previously described [22]. GLP-1 plasma concentrations were assessed by ELISA using a commercially available assay (EMD Millipore, Billerica, MA, USA).

### 2.4. Calculations and Statistical Analysis

Postprandial curves are presented as mean and area under the curve (AUC) with error bars showing the standard deviation. Fractions of a bile acid species were calculated as the sum of individual measurements of non-conjugated and glycine and taurine-conjugated forms. Total bile acids were calculated cumulatively by adding up all individual bile acid measurements. When bile acid levels were below the detection limit, plasma concentrations were set on 0.025 µmol/L to allow statistical analyses. The homeostasis model assessment for insulin resistance (HOMA-IR) and β cell function (HOMA-B) were calculated from fasting plasma glucose and fasting plasma insulin [23]. HOMA-IR and HOMA-B data were analysed using the unpaired Mann–Whitney U test. To understand insulin sensitivity during the meal, we calculated the cumulative Matsuda index at all time-points relative to baseline (t = 0 min) [24]. To analyse postprandial curves, a two-way repeated measures ANOVA followed by uncorrected Fisher’s Least Significant Difference test was performed using GraphPad Prism version 8.30 for Windows, GraphPad Software, La Jolla, CA, USA, www.graphpad.com (accessed on 26 October 2023).

## 3. Results

Table 1 shows the individual characteristics of the CTX patients and the age, sex and BMI-matched controls. Some of the CTX patient characteristics have been described before [25].

### 3.1. Postprandial Plasma Total Bile Acids

At baseline, lower fasting levels of total CA and DCA levels in CTX patients were observed (Figure 1). Postprandial total bile acids were threefold lower in CTX patients when compared to healthy volunteers, which was expected given the cessation of CDCA therapy (Figure 1A). Levels of total CA and total DCA were undetectable in some CTX patients. Postprandial CA and DCA levels were higher in healthy participants (Figure 1B,D). Total CDCA was still present in CTX patients despite the cessation of CDCA therapy for three weeks but was much higher in healthy participants, at time points 120, 150 and 180 min (Figure 1C). UDCA levels after the meal were not different between the two groups (Figure 1E). AUCs of postprandial total bile acids, CA and DCA were lower in CTX patients compared to healthy participants (Table 2). The AUC for total CDCA tended to be lower in the CTX patients (*p* = 0.053) (Table 2).

### 3.2. Postprandial Plasma Unconjugated and Conjugated Bile Acids

Baseline unconjugated CDCA showed no differences between the groups (Figure 2B), whereas baseline unconjugated DCA was lower in CTX patients compared to controls (Figure 2C). Increased levels of unconjugated CA and DCA were observed in the control group (Figure 2A,C). No differences between unconjugated CDCA and UDCA levels were found between the groups (Figure 2B,D). Glycine-conjugated bile acids demonstrated no deviations at baseline between CTX patients and controls (Figure 3A–D). The postprandial curves for glycine-conjugated CA, CDCA and DCA were generally lower in CTX patients (Figure 3A–C), but glycine-conjugated UDCA levels were higher in CTX patients (Figure 3D). Postprandial levels of unconjugated bile acids displayed the same differences for CA, DCA and UDCA as the glycine-conjugated bile acids (Figure 2A,C,D). Unconjugated forms of CA and DCA were reduced in the CTX patients compared to the controls, whereas UDCA and CDCA forms showed no variability (Table 2). Lower levels of glycine-conjugated CA and DCA were found in the CTX patients. AUCs for glycine-conjugated CDCA tended to be lower in CTX patients (*p* = 0.053) but AUCs for unconjugated CDCA were equal in both groups. Finally, taurine-conjugated DCA were lower in the CTX patients, and no differences were found in taurine-conjugated CA and CDCA. Taurine-conjugated UDCA levels were below the detection limit in the control group (Table 2).

### 3.3. Plasma Glucose, Insulin, GLP-1 and FGF19

Postprandial plasma glucose levels were higher at time point 75 min in the control group compared to CTX, whereas at time point 180 min levels were higher in CTX patients compared to control (Figure 4C). Fasting plasma insulin levels were not different between CTX patients and controls (69 ± 27 vs. 48 ± 24 mmol/L, *p* = 0.84) but were higher postprandial at time points 90, 120, 150, 180 and 240 min. HOMA-B tended to be lower in CTX patients; however, this was not the case for HOMA-IR (Figure 4A,B). Fasting plasma GLP-1 levels were comparable between CTX patients and controls (2.3 ± 1.1 vs. 3.5 ± 3.5 pmol/L, *p* = 0.41) but were higher at time points 60 and 240 min in CTX patients (Figure 4D). Total AUC for the postprandial plasma glucose profile did not vary between CTX patients and controls (Table 2). Total AUC for postprandial insulin levels were equal between CTX patients and controls (Table 2). The Matsuda insulin sensitivity index presented lower insulin sensitivity of CTX patients in comparison to controls early in and later in the meal test (Figure 4E), whereas total AUC of Matsuda insulin sensitivity index was not significantly different (Table 2). The curve of postprandial GLP-1 levels appeared higher in CTX patients which was statistically confirmed at 60 min (Figure 4F). AUCs for postprandial plasma GLP-1 profiles were not different between CTX patients and controls (Table 2). Postprandial FGF19 levels were at each time point significantly lower compared to controls (Figure 4G). In addition, AUCs for FGF19 levels were significantly lower in CTX patients compared to controls (Table 2).

## 4. Discussion

Here, we provide the first in-depth characterization of postprandial bile acid profiles, FGF19 and GLP-1 levels as well as glucose metabolism in the CYP27A1-deficient condition CTX and show profound alterations in postprandial bile acid profiles (absent elevations in CA and DCA species) and a blunted postprandial FGF19 response but intact postprandial GLP-1 responses in CTX patients as compared to healthy matched controls.

As stated in the introduction, CDCA has historically been used to stimulate FXR and decrease BA synthesis. Prolonged administration of CDCA is, in general, well-tolerated, but dose-related side effects are inevitable and can potentially result in elevated serum transaminases [26]. A two or three-fold increase in transaminases may not be related to liver injury, as described in the National Cooperative Gallstone study. This might be attributed to the hydrophobic feature of CDCA [27]. Based on clinical experience and minimal bile acid loss from the enterohepatic circulation, we reasoned that cessation of CDCA and statin therapy would result in reduced CDCA levels in the CTX patients. Indeed, after three weeks of therapy cessation, CDCA levels were generally lower and postprandial peaks were smaller when compared to the matched controls. As expected, conjugated forms were much higher and explained the differences in total CDCA concentrations between CTX patients and controls [28].

Under normal circumstances, CA is formed from its precursors via sterol 12-alpha-hydroxylase (CYP8B1) that is partly regulated by insulin [29]. CYP8B1 determines the ratio by which the primary bile acids CA and CDCA are formed. CA has the lowest FXR affinity of the primary and secondary bile acids followed by DCA [15]. In CTX patients, CA and its corresponding secondary bile acid DCA were hardly detected, which is in line with our hypothesis. It might be likely that the CDCA profile in CTX patients has the same FXR agonistic potential as the mixed CA-DCA-CDCA profile in controls. Since bile acid synthesis in CTX patients is disrupted, it can be expected that FGF19 responses are absent as we confirm here. Even the residual CDCA concentration after cessation of therapy was not sufficient to fully increase FGF19 levels. Under normal circumstances, bile-acid-induced FXR activation results in production of FGF19 and, in turn, regulates bile acid synthesis rate via negative feedback [30,31]. A complicating issue here is that FXR affinity is difficult to assess given the intranuclear localisation of the receptor. This is different for the transmembrane TGR5 where the affinity of different conjugated and unconjugated bile acids has been assessed in detail [32]. DCA has more affinity for TGR5 compared to CDCA followed by CA. DCA cannot be dehydroxylated in the human liver and, therefore, is an important constituent of the human bile acid pool [3]. The lack of DCA in the CTX patients suggests that TGR5 agonism of their total bile acid pool may be lower. The secondary lithocholic acid (LCA) is derived from CDCA and is generally very low in the peripheral as opposed to the portal circulation [33]. Because CDCA levels were lower in CTX patients this might have resulted in lower LCA-dependent differences concerning TGR5 activation.

Besides bile acid changes, we investigated the postprandial plasma glucose, insulin and GLP-1 responses to the meal. We found a different postprandial plasma glucose profile in our CTX patients compared to healthy participants which we think deserves attention. First, the different shape of glucose AUC, despite equal HOMA-B and HOMA-IR, might reflect different insulin sensitivity during the meal or unusual gastrointestinal transit [34]. GLP-1 slows down gastrointestinal transit, so it can be reasoned that lower bile acids normally diminish GLP-1 secretion, thereby speeding up intestinal transit. As an example, administration of CDCA has been found to increase circulating GLP-1 levels and reduce gastric emptying [35]. Unexpectedly, postprandial GLP-1 levels showed a higher peak and were not lower in CTX patients compared to controls. The explanation for this finding remains speculative; we and others have shown that supplementing bile acids to a meal may increase postprandial GLP-1 concentrations [31,35,36]. Also, cholecystokinin (CCK-) mediated gallbladder emptying showed increased GLP-1 secretion in humans [37,38]. As discussed above, CDCA and UDCA as found in the CTX patients are not the preferred TGR5 agonistic bile acids opposed to DCA. Moreover, total bile acids were lower in the CTX patients compared with controls. Nevertheless, CTX patients presented a higher GLP-1 response. It might be the case that CTX specific aberrant bile acids have high TGR5 activating potential, thereby inducing GLP-1 secretion from L cells. Indeed, the toxic intermediates, e.g., 7α-hydroxy-4-cholesten-3-one (C4), in CTX may affect FXR or TGR5 signaling [39]. Recent work on the liver × receptor (LXR) suggests that this might apply [40]. Thus, it was discovered that cholestanol has bioactive properties and regulates nuclear receptor function by activating LXR. It is worth mentioning that the largest proportion of the bile acid pool is within the enterohepatic cycle [33]. Therefore, it remains speculative to what extent toxic intermediates, including 7α-hydroxy-4-cholesten-3-one have affected GLP-1 secretion.

Higher GLP-1 levels coincided with a prolonged insulin response in the CTX patients reflective of decreased insulin sensitivity as calculated by the Matsuda insulin sensitivity index. The Matsuda index has been used in meals [41,42]. To this end, we calculated the index for all time points relative to baseline, creating a dynamic overview. Unfortunately, we cannot distinguish between hepatic and muscle insulin sensitivity in great detail. Nevertheless, CTX patients seem more insulin resistant compared to controls later in the meal. It is unknown to what extent the responsible gene CTX mutations in CYP27A1 directly affect glucose homeostasis. However, our data suggest that CTX patients have some degree of insulin resistance which may be attributed to changes in bile acid composition.

Our study has several limitations including a small sample size. CTX is a rare disorder, and participants that can give informed consent are even more scant. The small sample size and the inter-individual variation in bile acid measures may introduce statistical type 1 as well as type II errors. We chose to design a study imposing as little burden as possible on the CTX patients. Hence, we employed a single and simple test day without the use of stable isotopes or glucose and insulin clamps.

In conclusion, we provide a unique characterization of postprandial bile acid and FGF19 levels in CTX patients that are associated with changes in postprandial glucose and insulin profiles as well as postprandial GLP-1 responses.

## Figures and Tables

**Figure 1 nutrients-15-04625-f001:**
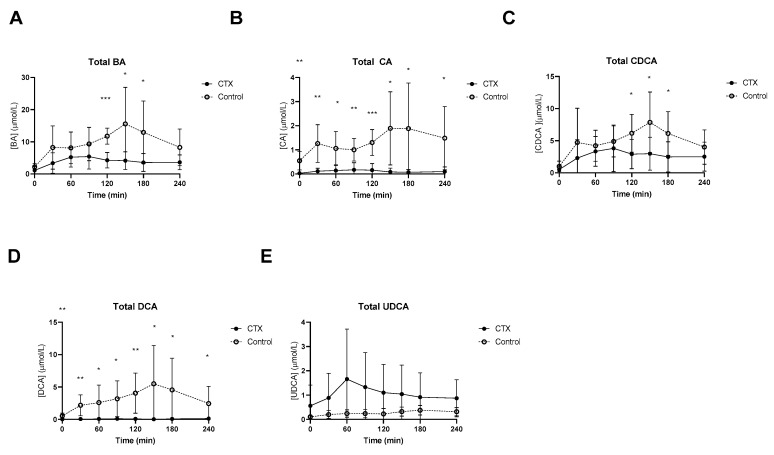
Postprandial levels of total bile acids (BA) (**A**), total cholic acid (CA) (**B**), total chenodeoxycholic acid (CDCA) (**C**), total deoxycholic acid (DCA) (**D**), and total ursodeoxycholic acid (UDCA) (**E**) in ● cerebrotendinous xanthomatosis (CTX) patients and ○ matched controls. *, ** and *** at the curve represents a significant effect on the time point (*p* < 0.05; *p* < 0.01; *p* < 0.001, respectively). Data are means ± standard deviation (SD).

**Figure 2 nutrients-15-04625-f002:**
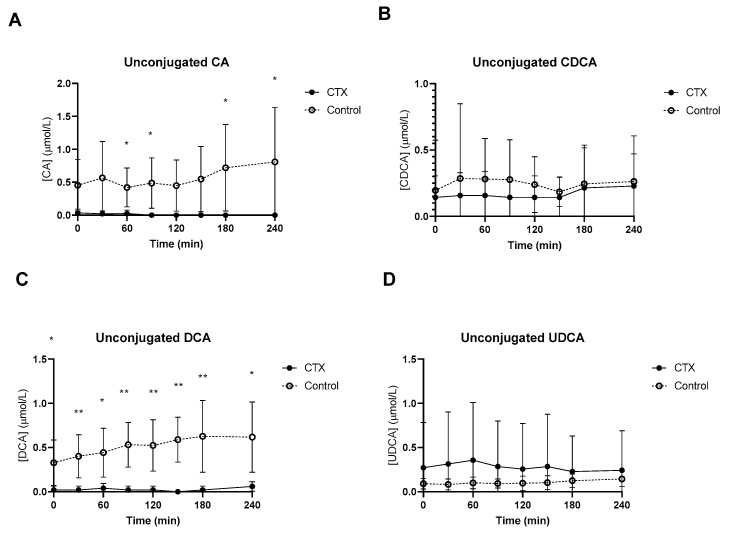
Postprandial levels of unconjugated cholic acid (CA) (**A**), chenodeoxycholic acid (CDCA) (**B**), deoxycholic acid (DCA) (**C**) and ursodeoxycholic acid (UDCA) (**D**) in ● cerebrotendinous xanthomatosis (CTX) patients and ○ matched controls. * and ** at the curve represents a significant effect on the time point (*p* < 0.05; *p* < 0.01). Data are means ± standard deviation (SD).

**Figure 3 nutrients-15-04625-f003:**
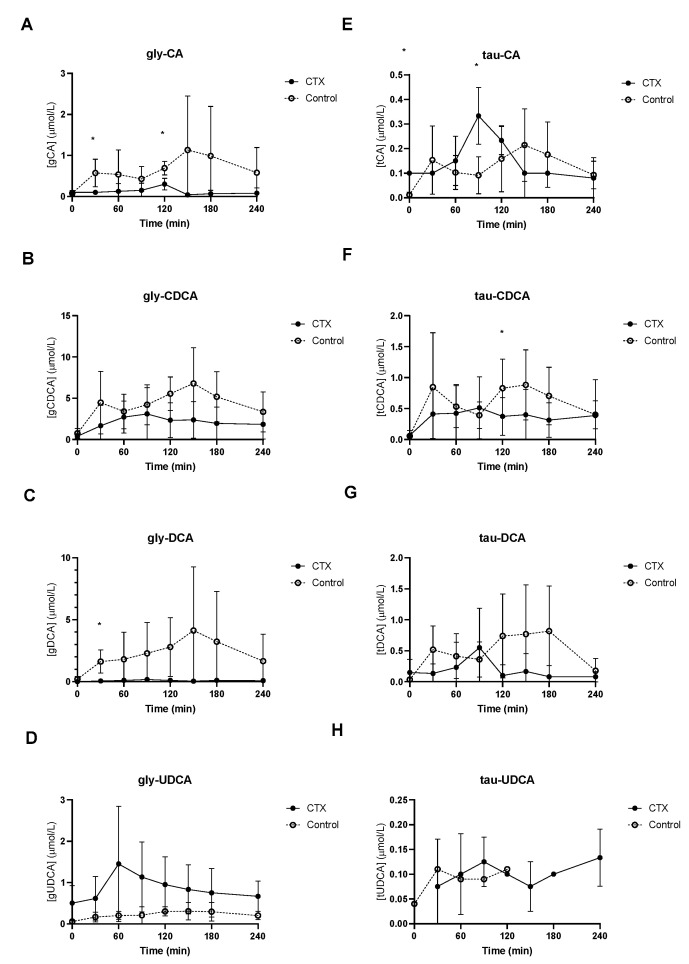
Postprandial levels of glycine-conjugated cholic acid (CA) (**A**), chenodeoxycholic acid (CDCA) (**B**), deoxycholic acid (DCA) (**C**) and ursodeoxycholic acid (UDCA) (**D**) and taurine-conjugated cholic acid (CA) (**E**), chenodeoxy cholic acid (CDCA) (**F**), deoxycholic acid (DCA) (**G**) and ursodeoxycholic acid (UDCA) (**H**) in ● cerebrotendinous xanthomatosis (CTX) patients and ○ matched controls. * at the curve represents a significant effect on the time point (*p* < 0.05). Data are means ± standard deviation (SD).

**Figure 4 nutrients-15-04625-f004:**
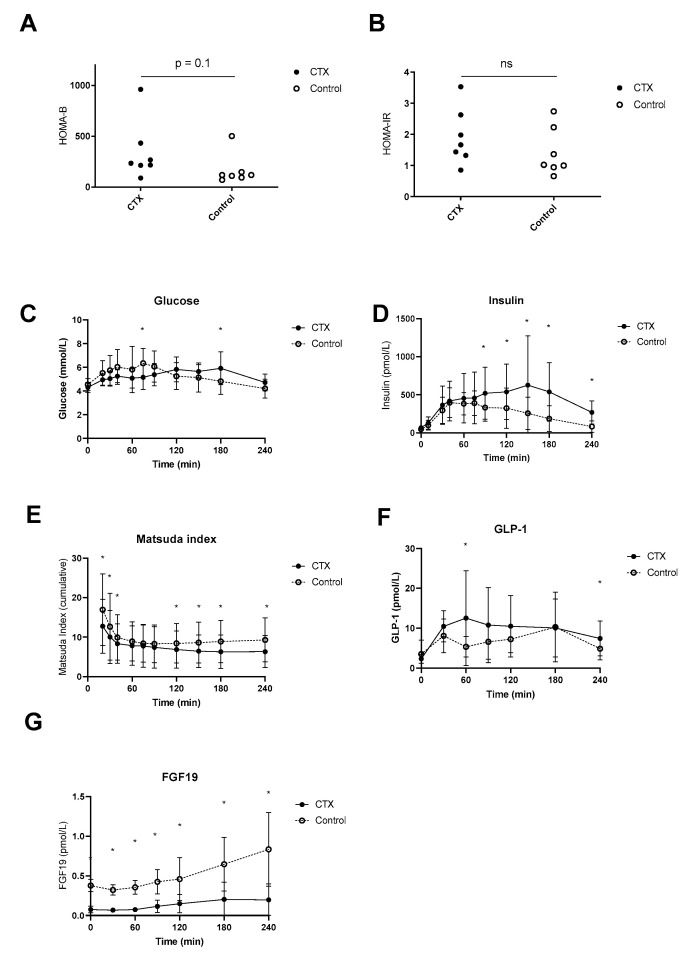
Postprandial hepatic insulin sensitivity (**A**), peripheral insulin sensitivity (**B**), excursion of glucose (**C**), insulin (**D**), Matsuda insulin sensitivity index (**E**), baseline-subtracted glucagon-like peptide 1 (GLP-1) (**F**) and fibroblast growth factor 19 (FGF19) (**G**) of ● cerebrotendinous xanthomatosis (CTX) patients or ○ matched controls. * at the curve represents a significant effect on the time point (*p* < 0.05). Data are means ± standard deviation (SD). ns: not significant.

**Table 1 nutrients-15-04625-t001:** Study population characteristics.

Participant	CTX Patients					Healthy Controls				
	Height (m)	Weight (kg)	BMI (kg/m^2^)	Age (years)	Gender (m/f) ^1^	Height (m)	Weight (kg)	BMI (kg/m^2^)	Age (years)	Gender (m/f) ^1^
1	1.82	75.0	22.6	43	m	2.00	95.0	23.8	40	m
2	1.85	70.0	20.5	40	m	1.85	65.0	19.0	42	m
3	1.68	76.0	26.9	46	f	1.71	59.7	20.4	44	f
4	1.58	83.0	33.2	45	f	1.57	63.6	25.8	51	f
5	1.80	97.0	29.9	51	m	1.90	107.2	29.7	49	m
6	1.65	72.5	26.6	52	m	1.85	96.4	28.2	62	m
7	1.66	78.0	28.3	53	f	1.65	68.0	25.0	54	f
Average	1.72	78.8	26.9	47		1.79	79.3	24.6	49	

^1^ m = male; f = female.

**Table 2 nutrients-15-04625-t002:** AUCs for postprandial glucose, insulin, GLP-1 and bile acid.

	CTX	Control	
	AUC ± SD	AUC ± SD	*p*-Value
Glucose (min × mmol/L)	1377 ± 113	1351 ± 244	0.44
Insulin (min × pmol/L)	4340 ± 2794	2833 ± 1469	0.23
Matsuda	1586 ± 272	2023 ± 326	0.08
GLP-1 (min × pmol/L)	2348 ± 1455	1690 ± 748	0.31
FGF19 (min × pmol/L)	34 ± 12	123 ± 22	<0.05
Total bile acids (min × μmol/L)	849 ± 526	2134 ± 839	<0.01
Total CA (min × μmol/L)	27 ± 33	282 ± 161	<0.001
Total CDCA (min × μmol/L)	581 ± 483	1095 ± 355	0.053
Total DCA (min × μmol/L)	17 ± 13	707 ± 562	<0.001
Total UDCA (min × μmol/L)	230 ± 252	54 ± 25	0.1
Unconjugated CA (min × μmol/L)	6 ± 2	112 ± 87	<0.001
Unconjugated CDCA (min × μmol/L)	36 ± 34	52 ± 47	0.48
Unconjugated DCA (min × μmol/L)	8 ± 4	108 ± 54	<0.001
Unconjugated UDCA (min × μmol/L)	62 ± 110	22 ± 13	0.35
Glycine conjugated CA (min × μmol/L)	16 ± 17	138 ± 108	<0.001
Glycine conjugated CDCA (min × μmol/L)	459 ± 417	950 ± 325	0.053
Glycine conjugated DCA (min × μmol/L)	13 ± 11	497 ± 464	<0.001
Glycine conjugated UDCA (min × μmol/L)	161 ± 142	64 ± 77	0.14
Taurine conjugated CA (min × μmol/L)	11 ± 20	32 ± 18	0.07
Taurine conjugated CDCA (min × μmol/L)	98 ± 79	145 ± 62	0.24
Taurine conjugated DCA (min × μmol/L)	2 ± 4	117 ± 95	<0.01
Taurine conjugated UDCA (min × μmol/L)	11 ± 8	nd	nd

Data were assessed with an unpaired *T*-test when data were normally distributed. 2 CTX: cerebrotendinous xanthomatosis; GLP-1: glucagon-like peptide 1; FGF19: fibroblast growth factor 19; CA: cholic acid; CDCA: chenodeoxycholic acid DCA; UDCA: ursodeoxycholic acid; AUC: area under the curve; SD: standard deviation; nd: not detected.

## Data Availability

The data presented in this study are available on request from the corresponding author. The data are not publicly available due to privacy of the patients.

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
