# Peer review of "Characterization of Postprandial Bile Acid Profiles and Glucose Metabolism in Cerebrotendinous Xanthomatosis"

_nutrients, 2023, doi:10.3390/nu15214625_

Round 1

Reviewer 1 Report

Comments and Suggestions for Authors

The article “Characterization of postprandial bile acid profiles and glucose metabolism in cerebrotendinous xanthomatosis” examined postprandial bile acid, glucose, insulin, GLP-1 and fibroblast growth factor 19 (FGF19) plasma profiles in patients with CTX and matched healthy controls. There are some shortcomings that need to be further improved.

Comments:

Q1. It is suggested to supplement references from the last three years

Q2. As stated, volunteers are seven patients and seven age, gender and body mass index matched controls. What is the average number of participants in this type of study that is statistically significant?

Q3. Line 90-92, Whether disturbances in bile acid synthesis and the enterohepatic circulation of bile acids as seen in CTX affect GLP-1 release and glucose metabolism remain unknown. Did the authors solved these questions though this paper?

Q4. Could bile acids affect glucose metabolism through FXR and TGR5 receptors?

Q5. Time (m), “m” means “min”?

Q6. As a key indicator of glucose metabolism, the results in Figure 3C are not significant.

Q7. The present study lacks sufficient research content as it primarily focuses on bile acid-related indicators, namely Figure 1, Figure 2, and Table 2, which are considered predictable indicators of CTX patients.

Q8. The core data of this paper should be represented by Figure 3, however, the relevant presentation is insufficient and the available data are inadequate to support the conclusions.

Author Response

Dear reviewer,

Thank you for reviewing our paper. Attached you can find our answers on your comments and questions.

With kind regards,

Soumia Majait

Reviewer 2 Report

Comments and Suggestions for Authors

The article is well founded and written and adds findings to the global knowledge of CTX.

Two minor issues are found. 1) The clinical synopsis in the Introduction has some bugs since cataracts are more common in early adulthood than in childhood and xanthomas are present only in 50% of cases. 2) The work analyses multiple hypothesis and it should be discussed why a multiple test correction was not considered in the statistics.

Author Response

(The authors gave the same response as above.)

Round 2

Reviewer 1 Report

Comments and Suggestions for Authors

No additional problems